# Evasion of host antioxidative response via disruption of NRF2 signaling in fatal *Ehrlichia*-induced liver injury

**Aditya Kumar Sharma[1], Abdeljabar El Andaloussi[1,2], Nahed Ismail[1]***

**1** Department of Pathology, College of Medicine, University of Illinois at Chicago, Chicago, Illinois, United States of America, **2** BioImmune Solutions Inc., 605–1355, Le Corbusier, Laval, Quebec, Canada

* ismail7@uic.edu

**Data Availability Statement:** All relevant data are within the manuscript and its Supporting Information files.

## Abstract

*Ehrlichia* is Gram negative obligate intracellular bacterium that cause human monocytotropic ehrlichiosis (HME). HME is characterized by acute liver damage and inflammation that may progress to fatal toxic shock. We previously showed that fatal ehrlichiosis is due to deleterious activation of inflammasome pathways, which causes excessive inflammation and liver injury. Mammalian cells have developed mechanisms to control oxidative stress via regulation of nuclear factor erythroid 2 related 2 (NRF2) signaling. However, the contribution of NRF2 signaling to *Ehrlichia*-induced inflammasome activation and liver damage remains elusive. In this study, we investigated the contribution of NRF2 signaling in hepatocytes (HCs) to the pathogenesis of *Ehrlichia*-induced liver injury following infection with virulent *Ixodes ovatus Ehrlichia* (IOE, AKA *E. japonica*). Employing murine model of fatal ehrlichiosis, we found that virulent IOE inhibited NRF2 signaling in liver tissue of infected mice and in HCs as evidenced by downregulation of NRF2 expression, and downstream target GPX4, as well as decreased NRF2 nuclear translocation, a key step in NRF2 activation. This was associated with activation of non-canonical inflammasomes pathway marked by activation of caspase 11, accumulation of reactive oxygen species (ROS), mitochondrial dysfunction, and endoplasmic reticulum (ER) stress. Mechanistically, treatment of IOE-infected HCs with the antioxidant 3H-1,2-Dithiole-3-Thione (D3T), that induces NRF2 activation, attenuated oxidative stress and caspase 11 activation, as well as restored cell viability. Importantly, treatment of IOE-infected mice with D3T resulted in attenuated liver pathology, decreased inflammation, enhanced bacterial clearance, prolonged survival, and resistance to fatal ehrlichiosis. Our study reveals, for the first time, that targeting anti-oxidative signaling pathway is a key approach in the treatment of severe and potential *Ehrlichia*-induced acute liver injury and sepsis.

## Author summary

*Ehrlichia* is a Gram-negative, obligate intracellular bacterium that causes the most prevalent life-threatening, tick-borne disease in the United States: human monocytic

**Funding:** This work was funded by the College of Medicine, UIC, Department of Pathology Start-up funds (N.I.). The funders had no role in the study design data collection and analysis, decision to publish, or preparation of the manuscript.

**Competing interests:** The authors have declared that no competing interests exist.

ehrlichiosis (HME). *Ehrlichia* infect mononuclear phagocytes, endothelial cells and hepatocytes. *Ehrlichia* target liver, which is the main initial site of infection and pathology, with hepatic injury and sepsis are main cause of death in HME. Liver damage in HME patients is associated with few organisms in blood and other tissues, suggesting that severity is not due to overwhelming infection, but rather to immunopathology. This study investigates the role of oxidative stress in *Ehrlichia*-induced liver injury. We found that virulent *Ehrlichia* disrupts anti-oxidative NRF2 pathway, leading to increase mitochondrial damage, production of reactive oxygen species, excessive inflammation and activation of deleterious innate immune signaling pathways such as inflammasomes. Using murine model of ehrlichiosis, we found that restoration of NRF2 signaling by treatment with antioxidant compounds ameliorate oxidative stress, inflammasome activation, liver pathology and protect mice against fatal ehrlichiosis-associated with sepsis. This study defines a novel immune-evasion mechanism where intracellular bacterium target anti-oxidative signaling response in hepatocytes to promote intracellular survival and dissemination, which in turn cause uncontrolled inflammation, liver damage, and sepsis.

## Introduction

Innate immune response is the first line of defense against harmful pathogens. At the core of this response are various cellular mechanisms that use oxidative stress to destroy invading pathogens. Mammalian cells have a balance between generation and elimination of reactive oxygen species (ROS); an imbalance in this homeostasis leads to oxidative stress. Regulated oxidative stress is beneficial to the host as it eliminates pathogens from the body. ROS are generated through mitochondrial oxidative phosphorylation in the mitochondria or from exogenous sources in peroxisomes and endoplasmic reticulum (ER) [1]. Increased ROS has been linked to various liver diseases, including nonalcoholic fatty liver disease (NAFLD), non-alcoholic steatohepatitis (NASH), characterized by mitochondrial dysfunction and metabolic reprogramming[2–4]. NRF2 is a transcription factor that counteracts the deleterious effect of increased ROS generation [5]. Under normal state, NRF2 is constitutively expressed in the cytoplasm of cells and associated with a repressor protein Kelch-like ECH-associated protein 1 (KEAP-1), which function to maintain low levels of free NRF2. In response to various stressors such as infection, NRF2 dissociates from KEAP1 and translocates to the nucleus to activate transcription of its downstream targets, including genes involved in glutathione and lipids metabolism, detoxifying and antioxidant pathways and mitochondrial function [5].

HME is the most prevalent emerging tick-borne disease characterized by non-specific flu-like symptoms and acute liver dysfunction [6,7]. If patients are not treated during initial stages of infection, HME can progress to life threatening complications such as multi-organ failure. The etiologic agent of HME is *Ehrlichia* that infects reticuloendothelial organs, primarily liver, lung, and spleen [6,7]. Unlike other gram-negative bacteria, *Ehrlichia* lack lipopolysaccharides (LPS) and peptidoglycans. Macrophages, hepatocytes (HCs), and endothelial cells are target cells for *Ehrlichia*[7–10]. The bacterium resides in phagosomes/endosomes that do not undergo lysosomal-endosomal fusion as evidenced by lack of lysosomal markers on endosomes/phagosomes containing *Ehrlichia* morulae [11,12]. Employing murine models of mild and fatal ehrlichiosis, we have shown that fatal ehrlichiosis caused by infection with virulent IOE is due to deleterious activation of inflammasome pathways, causing dysregulated inflammation, cell death and liver damage [7,10,13]. Inflammasome activation in macrophages during fatal infection is due to MYD88-mediated, mTORC1-dependent inhibition of autophagy

induction and flux, as well as defective mitophagy [13]. Interestingly, earlier studies have indicated that *E. chaffeensis*, major etiology of HME, alter mitochondrial metabolism and trigger mitochondrial damage [14].

In this study, we examined the impact of infection with virulent *Ehrlichia* (IOE) on NRF2 signaling pathway *in vivo* and in an *in vitro* infected HCs. Our data showed that IOE inhibit NRF2 signaling, as a possible immune evasion mechanism, that likely enable cell death and bacterial dissemination. Activation of NRF2 signaling using a small molecule D3T restored cellular function, attenuated inflammation and liver pathology as well as improved survival of infected mice following infection with virulent IOE.

## Material and methods

### Ethics statement

The animal experiments conducted in the study was approved by the University of Illinois at Chicago Animal Care and Committee and conducted in accordance with the local legislation and institutional requirements. All animal experiments were performed according to the guidelines of the American Association for the Assessment and Accreditation of Lab Animal Care (AAALAC).

### Mice and *Ehrlichia* infection

Female 7–8-week-old C57BL/6 were used in the experiments and obtained from Jackson Laboratory (Bar Harbor, ME). All the mice were maintained in a specific pathogen-free environment under a consistent light-dark cycle required for regulation and maintenance of the mice's circadian rhythms and behavior. The experiment conducted in the study was conducted under the Assurance Number 00290(A3460-01)) and IACUC protocol number (21–117) approved by the Institutional Animal Care and Use Committees, University of Illinois. Mice were intraperitoneal (i.p) injected with a high dose of highly virulent IOE *Ixodes ovatus Ehrlichia* (IOE, AKA *E. japonica*) and mildly virulent *Ehrlichia muris* (EM) inoculum of $10^3$ and $10^4$ bacteria/mouse for infection. All the experimental mice were monitored daily for signs of illness and survival. For *in vivo* intraperitoneal injection of D3T, mice were given with D3T, 10 mg/kg body weight consequently for first 5 days. All the mice were sacrificed at designated time points to harvest organs.

### Cell culture

Murine hepatocyte cell lines (HCs) AML12 (ATCC CRL2254) was used for *in vitro* experiments. HCs were grown and maintained as described [10]. IOE and EM bacteria were used for infection at the multiplicity of infection (MOI) of 1:10. HCs were treated with D3T at indicated doses, and then IOE was added two hours later. All the cells were collected at 24h post-infection (p.i.) for further analysis.

Other Material and Methods are included in S1 Text.

## Results

### Virulent *Ehrlichia* induces liver damage and fatal ehrlichiosis

We first evaluated the outcome of infection following infection with *Ixodes ovatus Ehrlichia* (IOE, AKA *E. japonica*) and *Ehrlichia muris* (EM) species, respectively. To this end, C57BL/6 mice were infected via intraperitoneal (i.p.) route with high doses of IOE and EM. On day 7 post infection (p.i.), liver sections from EM-infected mice had minimal cell death and inflammation with no evidence of steatosis compared to uninfected liver (Fig 1A and 1B). In contrast, liver sections from IOE-infected mice exhibited multiple foci of necrotic cells (Fig 1A and 1B),

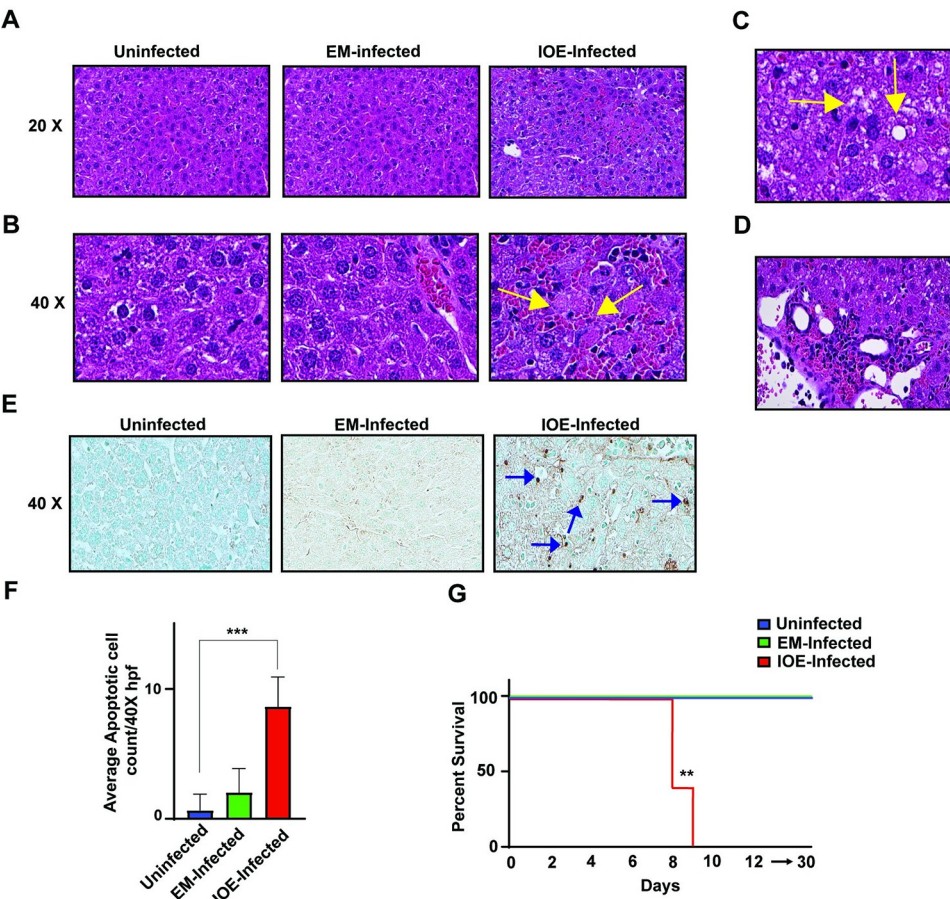

**Fig 1. Virulent *Ehrlichia* induces liver damage and fatal ehrlichiosis.** IOE-infected WT mice have altered liver pathology with increased hepatic necrosis and microvesicular steatosis. (A) Representative liver sections from uninfected, EM-infected and IOE-infected WT mice harvested on day 7 p.i. Liver sections were stained by hematoxylin and eosin (H&E). Liver from IOE-infected WT mice, but not EM-infected mice, exhibited marked necrosis (black arrows) (B), hepatic steatosis (C), and inflammatory cell infiltration composed primarily of lymphocytes (D). **(E)** Representative TUNEL staining of liver sections from uninfected, EM-infected WT and IOE-infected WT mice showing significantly higher number of brown- stained apoptotic hepatocytes (P < 0.001) in IOE-infected WT mice, compared with EM-infected mice. (F) Quantification of TUNEL positive HCs/ 40x hpf in uninfected, EM-infected, and IOE-infected Liver tissue. (G) Survival curve of EM-infected WT and IOE-infected WT mice (n = 9 mice/group), showing significant 100% survival of EM mice (P < 0.01) beyond day 30 compared to 100% mortality of IOE-infected mice on days 9–10 p.i. Data are expressed as means ± SD and are representative of one out of three independent experiments with 9 mice per group. *P ≤ 0.05, **P ≤ 0.01, ***P ≤ 0.001. Original magnification, x20 and ×40. HPF, high-power field.

hepatic steatosis (Fig 1C), and multiple inflammatory infiltrates (1D) on day 7 p.i. The number of apoptotic cells stained by TUNEL assay was significantly higher in IOE-infected liver tissues compared to EM-infected liver tissues (Fig 1E and 1F). All IOE-infected mice succumbed to infection between 8 to 12 days p.i. (i.e., 100% mortality), while all EM-infected mice developed mild and self-limited disease with 0% mortality (Fig 1D).

## Virulent *Ehrlichia* inhibits NRF2 nuclear translocation and activation in murine model of fatal ehrlichiosis

Studies have demonstrated the significance of antioxidant NRF2 signaling in prevention of hepatic necrosis and apoptosis associated with inflammation and liver injury [15–17]. We

hypothesized that IOE-induced liver damage and steatosis could be due to dysregulation of NRF2 signaling. To examine this hypothesis, we first examined the expression of NRF2 in the livers of IOE and EM-infected mice. Immunoblot analysis showed that IOE infection decreased expression of antioxidant NRF2 protein in the liver at day 7 p.i. when compared to uninfected and EM-infected liver (Fig 2A).

NRF2 regulates the expression of many downstream genes mediating different functions, such as NAD(P)H quinone oxidoreductase 1 (NQO1, enzyme that plays a role in redox balance), Glutathione peroxidase 3 (GPX3, enzyme involved in detoxification of hydrogen peroxide), Glutathione peroxidase 4 (GPX4, a phospholipid hydro peroxidase that prevent lipid peroxidation), and Thioredoxin reductase 1 (TXNRD1, an enzyme responsible for detoxifying toxins) (S1 Table) [5,18–29]. We measured the expression of GPX4 at protein level in EM- and IOE-infected mice. Compared to uninfected mice, EM infection did not change the expression level of total GPX4 in liver tissues (Fig 2B). In contrast, IOE-infected mice have a significant decrease in protein level of GPX4 in liver tissues compared to uninfected and EM-infected liver (Fig 2B).

Our results showed no significant changes in mRNA expression of these genes in the liver of EM-infected mice compared to uninfected mice. In contrast, IOE-infected liver tissues had significant lower mRNA expression of *nqo1* and *gpx4*, but not *gpx3*, (Fig 2C) when compared to uninfected and EM-infected mice. Surprisingly, the expression of *txnrd1* was markedly increased in IOE-infected liver tissues when compared to uninfected and EM-infected liver tissues (Fig 2C).

We next examined whether virulent IOE impedes the host antioxidant machinery by interfering with NRF2 translocation to the nucleus. To this end, we examined the subcellular location of NRF2 in the liver cells harvested from infected mice on day 7 p.i. by separating the cytoplasmic and nuclear fractions as described in Material & Methods. Appropriate controls for each fraction were included to exclude any possibility of cross-contamination. Notably, we found that infection with IOE resulted in a significant decrease in NRF2 protein expression in the nuclear fractions of liver lysates compared to uninfected mice (Fig 2D and 2E), suggesting an impaired NRF2 nuclear translocation. Additionally, NRF2 level in the total liver lysate as well as in the cytoplasmic fraction of IOE-infected liver lysate was significantly decreased compared to uninfected mice. Since NRF2 is known to be constitutively degraded prior to its translocation to nucleus, our data suggest that decreased total and cytoplasmic expression of NRF2 is due to proteasomal degradation and lack of stability of NRF2 following IOE infection. Further the nuclear translocation and expression of NRF2 and activation in uninfected mice were associated with expression of both nuclear and cytoplasmic GPX4 isoforms. In contrast, we only detected nuclear, but not cytoplasmic, GPX4 in IOE-infected liver lysates (Fig 2D and 2E). In mice and humans, distinct GPX4 isoforms with different subcellular localization are known to be produced through alternative splicing and transcription initiation; cytoplasmic GPX4, mitochondrial GPX4 (mGPX4), and nuclear GPX4 (nGPX4) [30]. The cytoplasmic GPX4 is the isoform essential for cell survival and prevention of lipid peroxidation and ferroptosis cell death [31,32]. Together, our data suggests that virulent IOE infection not only interferes with the nuclear translocation and activation of NRF2, but also with the stability of cytoplasmic NRF2, which is associated with significant downregulation of cytoplasmic GPX4 and antioxidant response.

## D3T restores NRF2 activation and its downstream signaling

Prior study from our lab indicated that hepatocytes are major target cells for *Ehrlichia*, and they play key roles in the pathogenesis of fatal ehrlichiosis [10,13]. To determine hepatocyte-specific response, we examined the NRF2 activation and signaling in IOE-infected murine

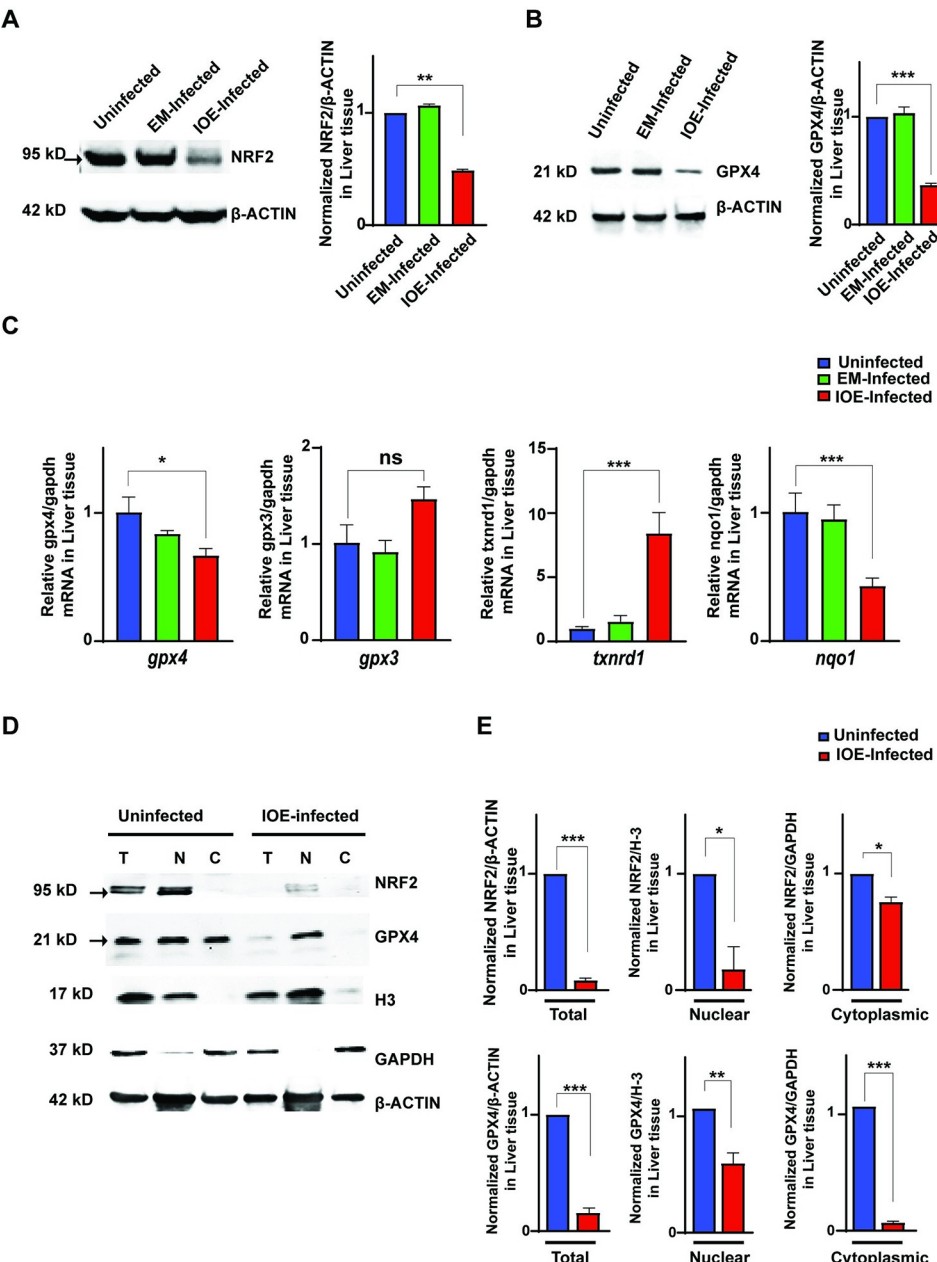

**Fig 2. Virulent *Ehrlichia* abrogates the expression and nuclear translocation of NRF2. (A, B)** Western blot showing NRF2 and GPX4 protein expression in the liver lysates from indicated groups at day 7 p.i. Data were normalized to β-actin as loading control. (**C**) mRNA expression of NRF2 downstream genes- *gpx4*, *gpx3*, *txnrd1*, and *nqo1* in the liver of indicated groups normalized to GAPDH. (**D, E**) Western blot showing expression of NRF2 and GPX4 in the total (T), nuclear (N) and cytoplasmic (C) fractions of liver cell lysates from uninfected and IOE-infected WT mice at day 7 p.i., and normalization to loading controls (β-actin, H-3 and GAPDH). Data are expressed as means ± SD and are representative of one out of three independent experiments with 9 mice per group. *P ≤ 0.05, **P ≤ 0.01, ***P ≤ 0.001.

hepatocyte cell lines (HCs). IOE infection decreased total protein expression of NRF2 in HCs compared to uninfected HCs (Fig 3A). To directly assess whether pharmacologic activation of NRF2 signaling in IOE-infected HCs restore expression of NRF2, we treated infected and uninfected HCs with different doses of D3T, a small molecule known to induce NRF2

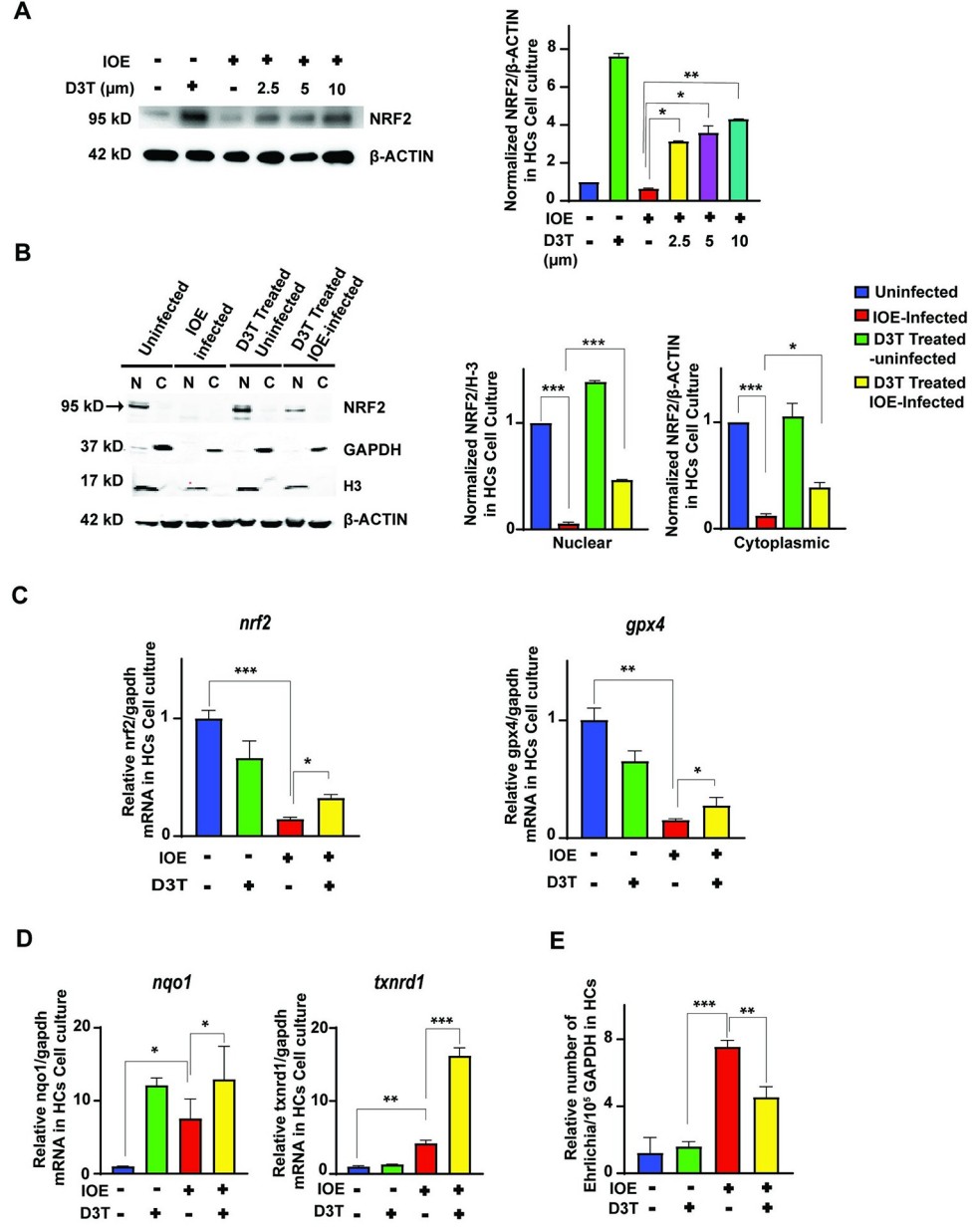

**Fig 3. The addition of D3T leads to an increase in the expression of NRF2 and its downstream target genes. (A)** Representative western blot showing the dose-dependent effect of D3T addition on NRF2 protein expression in infected HCs at 24hr p.i. β-actin is used as a loading control. D3T was added at 2.5 μm, 5.0 μm and 10 μm to HCs followed by IOE infection. **(B)** Western blot showing expression of NRF2 in the total (T), nuclear (N) and cytoplasmic (C) fractions of hepatocytes lysates from uninfected and IOE-infected HCs at 24hr p.i., and normalization to loading controls (H-3). The nuclear NRF2 is partially restored using 2.5 μm D3T. **(C, D)** mRNA expression of NRF2 and its downstream genes- *gpx4*, *txnrd1*, *nqo1* in uninfected and infected HCs in the presence or absence of D3T (2.5 μm). Results are normalized to GAPDH. **(E)** qRT-PCR showing number of intracellular IOE in uninfected and IOE-infected HCs cultured with or without D3T (2.5 μm) at 24hr p.i. Results shown are mean ± SD of one experiment with three replicate per condition and representative of two independent experiments (*P<0.05, **P<0.01, ***P<0.001).

activation [33,34]. Addition of D3T to IOE-infected HCs significantly increased NRF2 expression in a dose-dependent manner at 24hr p.i. when compared to control cells (untreated/IOE-infected, and uninfected cells treated with D3T) (Fig 3A). In all subsequent experiments, D3T

was used at a concentration of 2.5 μm, unless otherwise indicated. To determine the subcellular localization of NRF2 following D3T treatment, we isolated nuclear and cytoplasmic fractions from infected and uninfected cells harvested at 24hr p.i., and measured NRF2 expression. The NRF2 expression level in cytoplasmic and nuclear fractions was normalized to β-actin and H3, respectively. Consistent with *in vivo* data, IOE infection resulted in significant decrease in activation of NRF2 as evidenced by substantial lack of NRF2 expression in the nuclear fraction of HCs when compared to uninfected cells (p<0.001) (Fig 3B). Notably, we did not detect NRF2 expression in the cytoplasmic fraction of IOE-infected cells, suggesting that cytoplasmic NRF2 may undergo proteasomal degradation at that time point. Importantly, treatment of IOE-infected HCs with D3T resulted in significant NRF2 nuclear translocation (p<0.001) as marked by increased expression of nuclear NRF2 compared to infected but untreated HCs (Fig 3B).

Addition of D3T to IOE-infected HCs also increased mRNA expression levels of *nrf2* and *gpx4* when compared to controls (Fig 3C). The mRNA expression of *nqo1* was significantly increased upon the addition of D3T (Fig 3D). Like *in vivo* data, expression of *txnrd1* was increased upon IOE infection, and that increase was further enhanced upon treatment with D3T (Fig 3D). Importantly, analysis of *Ehrlichia* 16s rDNA by RT-PCR indicated that restoration of NRF2 activation in IOE-infected HCs by D3T resulted in decreased number of intracellular IOE in infected cells at 24hr p.i. compared to untreated/infected HCs (Fig 3E). Together, these findings suggest that *Ehrlichia* manipulates the host's antioxidant defense mechanism to induce liver damage and evade anti-bacterial host response by inhibiting activation of NRF2 signaling pathway in HCs.

## Impaired NRF2 activation in HCs during IOE infection causes mitochondrial dysfunction

We have previously shown that virulent IOE induced mitochondrial dysfunction in macrophages, which is associated with deleterious inflammasome activation [13]. We hypothesized that IOE-induced inhibition of NRF2 signaling in HCs is linked to mitochondrial dysfunction. To evaluate this hypothesis, we first measured mitochondrial transmembrane potential in infected HCs, in the presence or absence of D3T, using JC-1 dye and flow cytometry. JC-1 monomers aggregate on the energized and healthy mitochondria and form red fluorescent J-aggregates. By contrast, JC remains in the monomeric form in unhealthy or apoptotic cells with low mitochondrial potential, which show as green fluorescence. Thus, a decrease in red/green fluorescence intensity ratio is a sign of mitochondrial depolarization. Our data showed that IOE induces depolarization of mitochondria in infected HCs at 24h p.i., where 34% of cells exhibit low membrane potential (green, fluorescent) compared to 4% in uninfected cells (Fig 4A). Treatment of IOE-infected cells with D3T restored the mitochondrial potential in IOE-infected cells in a dose-dependent manner with ~ 21% and 3% (10-fold lower than untreated/infected cells) of cells exhibit low membrane potential at 2.5 μm and 5.0 μm, respectively (Fig 4A and 4B). Enhanced mitochondrial function in IOE-infected, D3T-treated cells coincide with ~ 3-fold decreased mitochondrial ROS, as measured by mitosox staining and flow cytometry, when compared to IOE-infected, D3T-untreated HCs (7% mitosox positive cells vs 20%) (S1A Fig). Similarly, the mean fluorescence intensity (MFI) of intracellular mitochondrial ROS measured by mitoSox staining and flow cytometry was significantly decreased in D3T treated and infected cells compared to untreated cells, although we have not detected a dose-response effect (S1B Fig). Additionally, total ROS production by IOE-infected HCs was significantly higher than uninfected cells, and that increase was attenuated upon addition of D3T (Fig 4C). D3T-induced attenuation of ROS correlated with increased viability of IOE-infected HCs (Fig 4D).

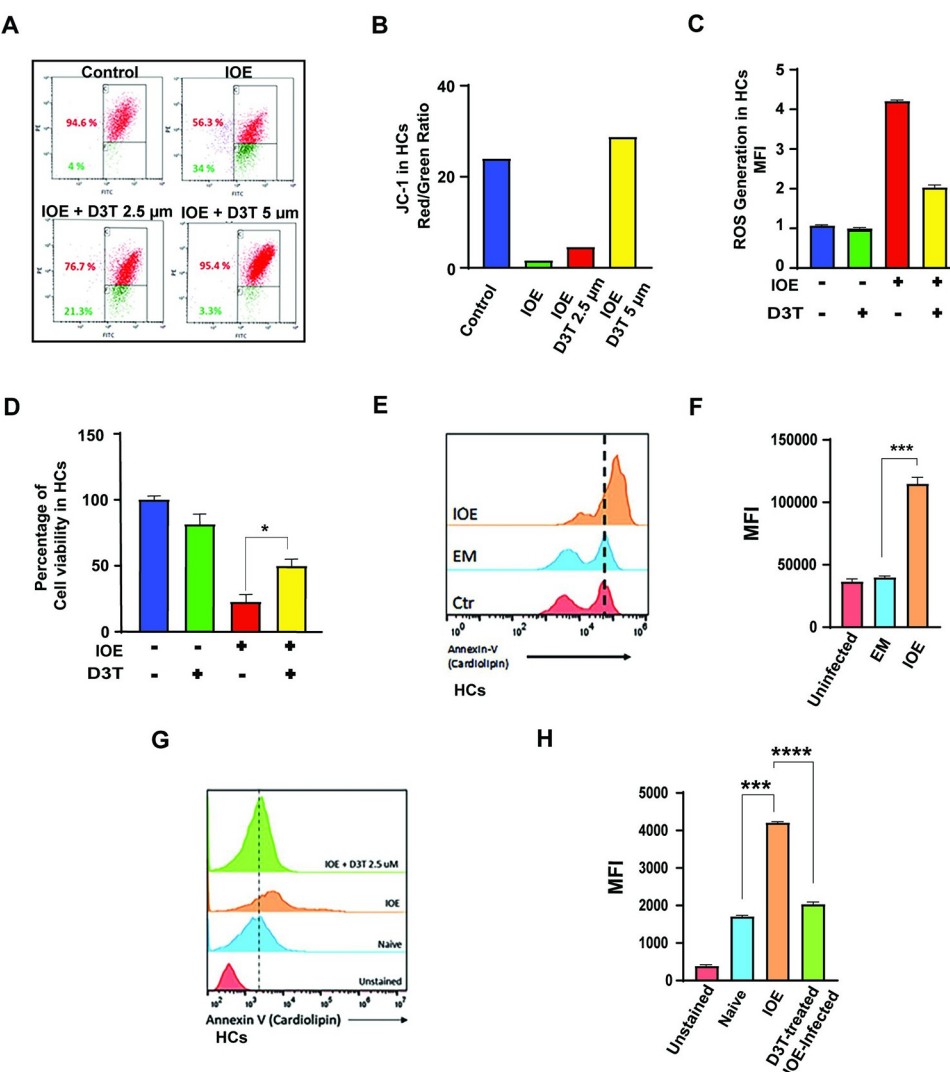

**Fig 4. Restoration of mitochondrial functioning and cell viability in IOE-infected HCs upon Nrf2 induction. (A)**
Dot blot-flow cytometry analysis of mitochondrial membrane potential in uninfected and infected HCs cultured with
or without D3T at concentration of 2.5 μm, at 24hr p.i. using JC dye. **(B)** The ratio of healthy (red staining) to
unhealthy (green) mitochondrial in indicated cell culture conditions. **(C)** Total intracellular ROS generation in HCs is
measured using Cellular ROS assay kit. Uninfected, IOE-infected, and D3T-treated IOE-infected HCs are used for
ROS formation. D3T (2.5 μm) is used in the experiment **(D)** Cell viability in indicated groups in presence and absence
of D3T (2.5 μm) **(E, F)** Histogram and MFI analysis of annexin staining of oxidized cardiolipin on isolated
mitochondria from indicated culture conditions in presence or absence of D3T (2.5 μm) **(G, H)** Histogram and MFI
analysis of annexin staining of oxidized cardiolipin on isolated mitochondria from uninfected, and IOE infected HCs
cultured with or without D3T (2.5 μm). Results shown are mean ± SD of one experiment with three replicate per
condition and representative of two independent experiments (*P<0.05, **P<0.01, ***P<0.001).

To further assess mitochondrial function and integrity, we analyzed oxidized mitochondrial
cardiolipin (CL) using annexin staining and flow cytometry. CL is a major membrane phos-
pholipid that is only found in inner mitochondrial membrane and becomes externalized and
oxidized with cytochrome c oxidase because of mitochondrial oxidative stress. Mitochondria
were isolated from uninfected and IOE-infected HCs as described in Material and Methods,
and stained with annexin-V. Our results showed a higher percentage and MFI of annexin posi-
tive mitochondria from IOE-infected HCs compared to uninfected HCs and EM-infected HCs
(Fig 4E and 4F). Treatment of IOE-infected cells with D3T significantly decreased percentage

and MFI of annexin positive mitochondria (oxidized CL) to a level that is similar to the level of annexin positive mitochondria in HCs infected with avirulent EM (Fig 4G and 4H). Using immunofluorescence and staining with Mito Tracker-Red, we found that IOE-infected HCs had a higher number of mitotracker red stained cells compared to uninfected cells (S1C Fig). However, this increase in the number of mitochondria did not seem to be accompanied by a change in mitochondrial mass, as measured by measurement of mitochondrial DNA (S1D Fig). The above changes in mitochondrial functions and ROS production were associated with significantly enhanced survival and cell proliferation (S1E Fig) of IOE-infected HCs upon D3T treatment when compared to untreated infected HCs. Together these data confirm that D3T-mediated increased NRF2 expression during infection with virulent *Ehrlichia* restore mitochondrial structure, dynamics, and function as well as viability of HCs.

## Mitochondrial dysfunction in IOE-infected HCs is linked to dysregulation of PINK1 & PARKIN

Studies have shown that PINK1 and PARKIN play a vital role in maintaining the mitochondrial quality control pathway. PINK1 is a mitochondrial-targeted serine-threonine kinase and Parkin is a RING-between-RING (RBR) type E3 ubiquitin (Ub) ligase. Damage of mitochondria triggers stabilization of PINK1 on the mitochondrial outer membrane, and activation of PARKIN E3 ligase activity, targeting damaged mitochondria for degradation via mitochondrial autophagy (i.e., mitophagy). Thus, the cellular accumulation of PINK1 and PARKIN on mitochondria is a specific cellular response to mitochondrial damage (S1 Table) [35–41]. We thus measured the kinetics of expression of PINK1 and PARKIN in the liver lysates from mice infected with virulent IOE on days 3 and 7 p.i. Compared to uninfected mice, we detected a substantial decrease in the mRNA expression of *pink1*, *parkin* and *erp44* on day 7, but not at day 3 pi. (Fig 5A). We also detected a decrease in mRNA expression levels in *pink1*, *parkin* and *erp44* only in the IOE-infected liver tissues (Fig 5B). At the protein level, we also detected a decreased expression of PINK1 and PARKIN in IOE-liver lysate compared to uninfected or EM-infected liver lysates (Fig 5C and 5D). These data suggest that IOE-induced mitochondrial damage may be linked to inability of HCs to eliminate damaged mitochondria via mitophagy as a result of reduced expression of PINK1 and PARKIN.

## Virulent *Ehrlichia* upregulates three arms of unfolded protein response and downstream effector signaling pathways

Unfolded Protein Response (UPR) is a cellular stress response of the ER. Although UPR is one of the mechanisms by which ER restores cellular homeostasis, this mechanism is unable to alleviate ER stress in the context of inflammation and fatty accumulation (S1 Table) [42–48]. We hypothesized that oxidative stress during fatal ehrlichiosis induces ER stress and UPR in HCs, which may promote lipid accumulation and further tissue damage. To assess this hypothesis, we first examined mRNA expression of ER stress genes in the livers of IOE (fatal) and EM (non-fatal) infected mice. Our results showed a significant increase in mRNA expression of *perk*, *ire1*, *atf6*, *xbp1*, *xbp1s*, and *chop* in IOE-infected livers compared to uninfected and EM-infected mice (Fig 6A). Further, there was also a significant increase in mRNA expression of genes responsible for targeting misfolded proteins such as ERAD-related genes—*edem*, *gadd34*, and *dr5* in IOE-infected livers compared to controls (Fig 6A). We did not detect changes in ER stress genes in the livers of EM-infected mice, except for a slight increase in expression of IRE1 compared to uninfected mice (Fig 6A). Further, transmission electron microscopy (TEM) analysis of liver sections demonstrated a substantial distortion and disorganization of ER in the livers of IOE-infected and *E. muris* infected mice compared to uninfected

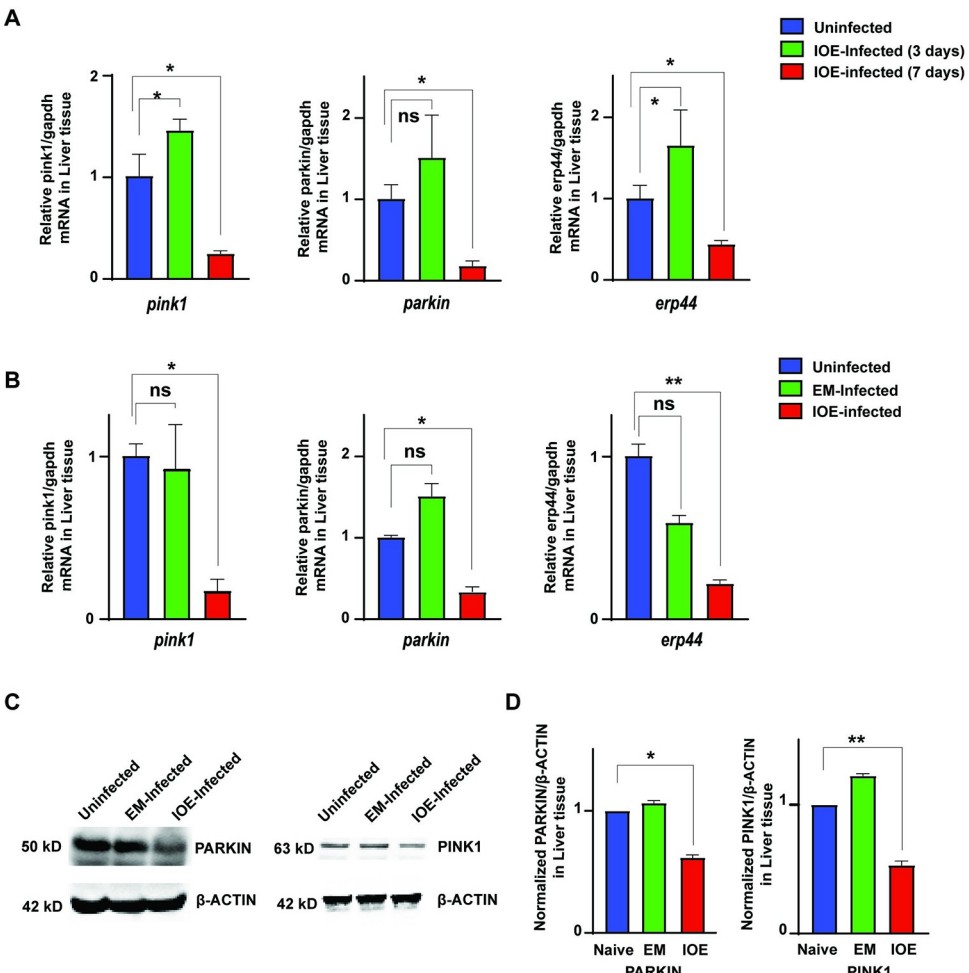

**Fig 5. Dysregulation of Mitochondrial genes in *Ehrlichia* IOE-infected liver in mice. (A)** mRNA expression of gene *pink1*, *parkin*, *erp44* at days 3 and 7 p.i. from liver lysate of IOE-infected WT mice. **(B)** mRNA expression of genes regulating mitochondrial dysfunction- *pink1*, *parkin*, *erp44* normalized to *gapdh* in liver cells from indicated mice groups. **(C, D)** Representative western blot and its analysis showing protein expression level of PINK1, PARKIN, and β-actin as loading control in the whole liver lysate of uninfected, EM-infected, and IOE-infected WT mice. Results shown are mean ± SD of one experiment with three mice/group and representative of three independent experiments. (*P<0.05, **P<0.01, ***P<0.001).

mice. However, mitochondrial cristae and structure was intact in EM-infected liver tissues. On the other hand, IOE-infected liver exhibited dysfunctional mitochondrial with abnormal mitochondrial morphology and disorganization with matrix swelling and collapsed cristae, suggesting mitochondrial dysfunction (Fig 6B). Further *in vitro* expression of *perk*, *ire1*, *xbp1* and *chop* in IOE-infected HCs was significantly upregulated compared to uninfected cells (Fig 6C).

Next, we examined the link between NRF2 signaling and ER stress in HCs following IOE infection. HCs were infected with IOE, in the presence or absence, of TUDCA (Tauroursodeoxycholic Acid), a chemical that attenuates ER stress [49,50]. As positive control, the expression of NRF2 was increased upon stimulation of uninfected HCs with PMA. Notably, Inhibition of ER stress by TUDCA in IOE-infected HCs resulted in a significant increase in the expression of NRF2 compared to untreated/infected cells (Fig 6D and 6E). These data suggest that IOE-induced inhibition of NRF2 expression and/or activation may be partly mediated by ER stress during severe *Ehrlichia* infection.

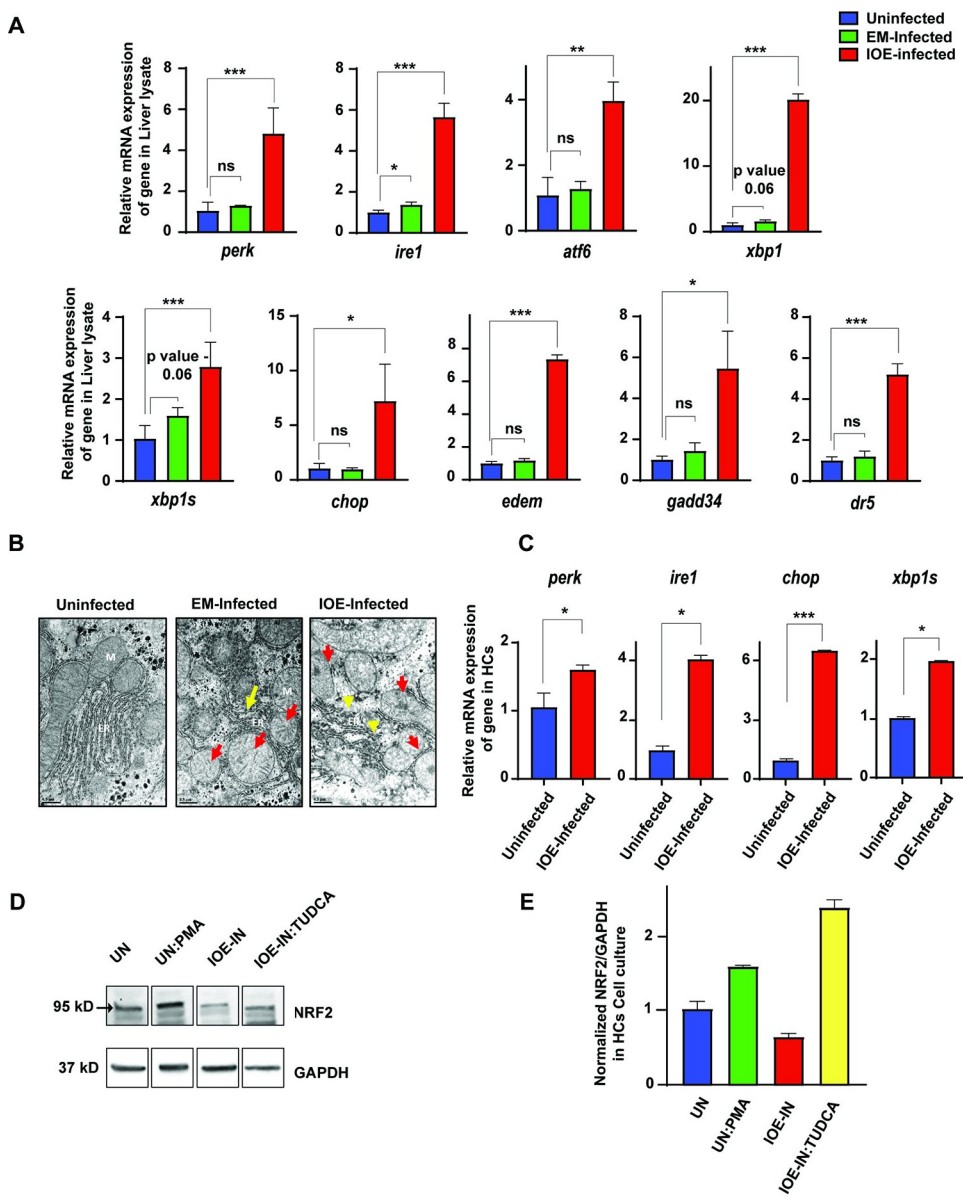

**Fig 6. *Ehrlichia* upregulates three arms of unfolded protein response and downstream effector signaling pathways. (A)** Graph showing mRNA expression data of ER-related genes- *perk*, *ire1α*, *atf6*, *XBP1*, *xbp1s*, *chop*, *edem*, *gadd34*, and *dr5* in whole liver lysate from uninfected, EM-infected, IOE-infected mice. **(B)** TEM images show the ER's condition in uninfected, EM-infected, and IOE-infected HCs. **(C)** mRNA expression of genes responsible for UPR- *perk*, *ire1α*, *chop*, and *xbp1s* in uninfected and IOE-infected HCs. **(D, E)** Representative western blot showing protein expression level of NRF2 and GAPDH as loading control in the HCs of indicated *in vitro* culture conditions. Protein samples were run in the same gel and the density of the bands was quantified from the same membrane. Results shown are mean ± SD of one experiment with three replicates/condition and representative of two independent experiments. (*P<0.05, **P<0.01, ***P<0.001).

## Restoration of active NRF2 signaling protects mice from fatal *Ehrlichia*-induced liver damage and sepsis

To investigate the impact of D3T treatment on the outcome of infection following IOE infection, we infected mice with a lethal high dose of IOE and treated mice with D3T from day 3–7 p.i. Intriguingly, treatment of IOE-infected mice with D3T prolonged survival whereas

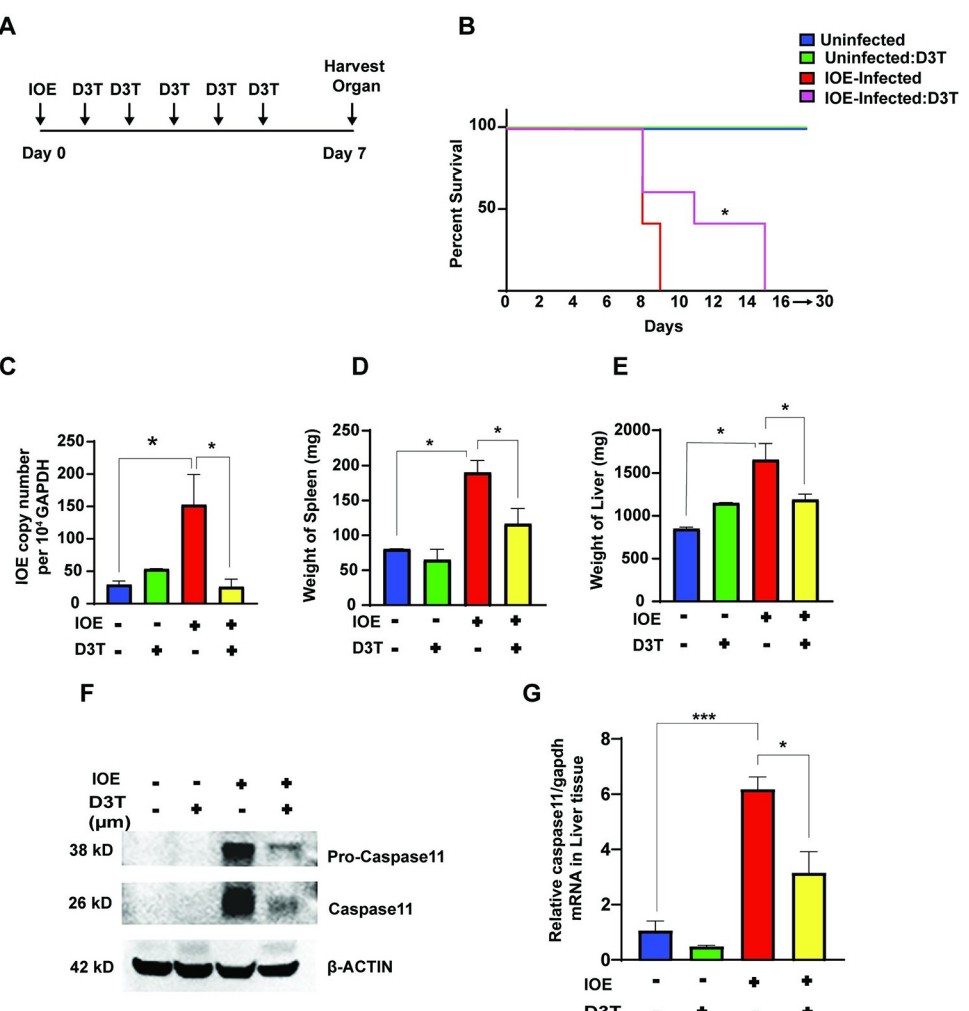

**Fig 7. *In vivo* impact of D3T on survival, inflamation, and bacterial burden. (A)** Experimental design describing timeline for administration of D3T. **(B)** Survival of indicated mice groups (n = 9/group) showing better survival upon D3T treatment. **(C)** Bacterial burden in the whole liver lysate in the uninfected, D3T-treated uninfected, IOE-infected, and D3T-treated IOE-infected. **(D, E)** Weight of spleens and livers from indicated mice groups. **(F)** Western blot showing protein expression of caspase 11 in the whole liver lysate. β-actin is used as a loading control. **(G)** mRNA expression of *caspase 11* in whole liver lysate from uninfected, D3T-treated uninfected, IOE-infected, D3T-treated IOE-infected cells. The mRNA expression is normalized to *gapdh*. Results shown mean ± SD of one out of three independents experiments with similar results. Number of mice/groups = 9. *P<0.05, **P<0.01, ***P<0.001.

approximately 40% of mice survived till day 15 p.i., while untreated infected mice succumbed to infection between 7–9 days p.i. (Fig 7B). Treatment of uninfected mice with D3T did not significantly influence liver weight when compared to uninfected and untreated mice (Fig 7E). Infection of mice with IOE significantly (p< 0.05) increased liver weight compared to unin-fected mice (Fig 7E). Notably, attenuated mortality in D3T treated, IOE-infected mice were associated with significant decrease in the weight of the livers and spleens, decreased total number of splenocytes, as well as decreased bacterial burden when compared to untreated controls (Fig 7C and 7E) (S1F Fig). These results suggest that D3T treatment restores the anti-oxidative response enhances bacterial clearance, and prevents hepatomegaly following fatal IOE infection.

## Activation of NRF2 signaling prevents activation of deleterious inflammasomes and protect mice from fatal ehrlichiosis

We next examined the impact of NRF2 signaling on inflammasome activation in IOE-infected HCs. Consistent with our prior finding, IOE infection stimulates activation of caspase 11 in liver tissue compared to uninfected liver tissue. In contrast, treatment of IOE-infected mice with D3T not only decreased activation of caspase 11, but also decreased protein expression of pro-caspase 11, suggesting that IOE-mediated, NRF2-dependent oxidative stress promote caspase 11 activation as well as regulate pro-caspase 11 expression (Fig 7F and 7G). Inhibition of caspase 11 activation in IOE-infected liver tissue upon D3T treatment correlated with increased cellular proliferation at 24hr p.i. compared to controls (S1E Fig), re-enforcing our prior data showing caspase 11-dependent death of HCs following IOE infection. Importantly, D3T treated and infected mice that survived fatal infection had marked attenuation of liver pathology at day 7 p.i., as demonstrated by H&E and TUNEL staining of liver sections (Fig 8A and 8B). Liver tissues from IOE-infected mice exhibited several foci of necrotic and apoptotic cells, as well as accumulation of lipid droplets consistent with hepatic steatosis. In contrast, liver tissues from D3T treated, IOE-infected mice had fewer fatty changes, less necrosis and apoptosis of HCs as well as cells lining liver sinusoid and blood vessels including endothelial cells and macrophages (Fig 8A and 8B). Quantitative analysis also showed that D3T treated mice have lower apoptotic cell count of kupffer and hepatocytes cells compared to IOE-infected mice p.i. 7-day infection (S1G Fig). Taken all together, restoration of NRF2 activation during severe *Ehrlichia* infection prevent liver injury and steatosis as well as protect mice against fatal sepsis.

## Discussion

Oxidative stress is considered as an imbalance between generation of ROS and oxidants and counteracting activity of antioxidants [51,52]. The role of antioxidant host defense system against *Ehrlichia* infection has not been studied. Fatal ehrlichiosis in mice and humans is associated with development of inflammation, hepatic steatosis, as well as inflammatory cell death [6,7]. In this study, we demonstrated that IOE trigger oxidative stress and liver damage via inhibition of NRF2 expression, which causes liver steatosis, accumulation of ROS, mitochondrial damage, ER stress and upregulation of unfolded protein response, as well as activation of deleterious non canonical inflammasome pathway marked by caspase 11 activation. Our data are consistent with findings in other non-infectious model systems showing the importance of NRF2 in regulating lipid peroxidation and fatty liver diseases [19,53,54].

Importantly, our study demonstrated an efficacy of small molecule D3T in prevention of fatal outcome and attenuation of liver pathology in preclinical animal model of HME. D3T is known to enhance a host's antioxidative and anti-inflammatory response by upregulating NRF2 and its downstream signaling. D3T treatment of HCs infected with virulent IOE, resulted in upregulation of NRF2 expression, NRF2 mediated expression of downstream antioxidants, reduction of ROS, improved mitochondrial function and structure, improved cell viability, and attenuated ER stress. Among antioxidant markers, we found that fatal IOE infection decreases expression of key molecules involved in several cellular processes including GPX4, TXNRD1 and NQO1 (S1 Table) [21,22,25,28]. S1 Table includes a summary of all differentially regulated genes and encoded proteins in fatal ehrlichiosis, which are known to play critical roles in multiple cellular process including oxidative responses, ER stress, autophagy, mitophagy, inflammation and cell death. As we indicated in S1 Table, TXNRD1 belongs to the family of thioredoxin reductases, which detoxify bacterial toxins, electrophilic compounds, environmental toxins, and reactive intermediates. TXNRD1 is also regulated by transcription

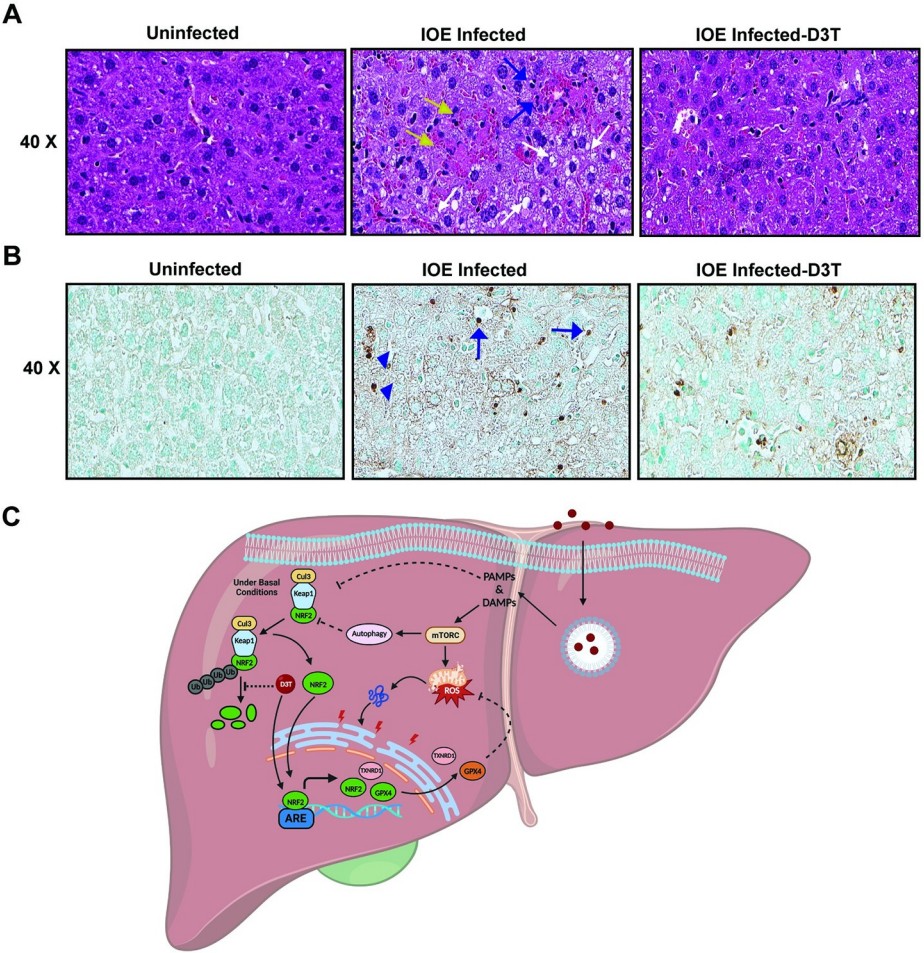

**Fig 8. D3T-induced activation of NRF2 protects mice against fatal ehrlichiosis. (A, B)** Representative H&E staining of liver sections from uninfected, IOE-infected WT, and D3T-treated IOE-infected WT mice on day 7 post-infection. Blue arrows indicate inflammatory infiltrates cells, white arrows indicate steatosis and yellow arrows indicate necrosis. We observed increased steatosis in the IOE-infected liver tissue in the histological images only and there is a marked decrease in steatosis, infiltrates inflammatory cells and necrosis upon D3T treatment. **(B)**. Representative TUNEL staining of liver sections from uninfected, IOE-infected WT, and D3T-treated IOE-infected WT mice on day 7 post-infection. The number of TUNEL-positive Kupffer cells and damaged HCs cells is higher in IOE-infected WT mice than in D3T-treated IOE-infected WT mice (refer S1G Fig). TUNEL-positive Kupffer cells and hepatocytes are indicated using arrows and arrowhead, respectively. **(C)**. A graphic model summarizing the signaling pathways in HCs during fatal *Ehrlichia* infection. Virulent *Ehrlichia* invades HCs, replicates inside the early endosomes without fusing with lysosomes, and secretes PAMPs into cytoplasm. Infection causes accumulation of intracellular ROS and mitochondrial damage and release of mitochondrial DAMPs into cytoplasm. Cytoplasmic PAMPs and DAMPs trigger activation of deleterious non canonical inflammasome pathway marked by caspase 11 activation. This process is accompanied by activation of mTORC1 activation, inhibition of NRF2 translocation to nucleus, via stabilizing KEAP1, which impair transcription of genes involved in antioxidant defense, detoxification, lipid metabolism, and inflammation. This further leads to increased ROS accumulation and ER stress in the HCs, and further amplification of inflammasome activation causing excessive inflammation and cell death. D3T treatment abrogates all these deleterious events via blocking proteasomal degradation of NRF2 and promoting its cytoplasmic translocation and activation, leading to protection against fatal *Ehrlichia*-induced liver injury. Abbreviations- PAMPs—Pathogen-associated molecular patterns, DAMPs—Damage-associated molecular patterns, mTORC—Mammalian target of rapamycin, ROS—Reactive oxygen species, Cul3—Cullin3, Keap1—Kelch-like ECH-associated protein 1, NRF2—Nuclear factor erythroid 2-related factor 2, Ub–Ubiquitin, D3T - 3H-1,2-Dithiole-3-thione, ARE—Antioxidant response element, TXNRD1—Thioredoxin reductase 1, GPX4—Glutathione peroxidase 4. Created with BioRender. com.

factors other than NRF2 [55]. The finding that TXNRD1 is highly upregulated while other NRF2 downstream gene targets such as GPX4 and NQO1 are downregulated suggest that TXNRD1 expression is partially dependent on transcriptional activation signal by NRF2 and other signaling pathway such as USF2 mediated upregulation of TXNRD1 may play a role in this disease process [20,55]. It is possible that the substantial upregulation of TXNRD1 in liver tissues of IOE-infected mice occurs as a counter-protective detoxification host response to ameliorate tissue damage and high bacterial burden.

Previous studies have demonstrated that GPX4 deficiency in mice leads to increased susceptibility to infection with *Mycobacterium tuberculosis*, as evidenced by increased lung necrosis and bacterial burden [21]. Additionally, GPX4 deficiency in macrophages has been linked to enhanced necrosis, which can be reversed by treatment with lipid peroxidation inhibitor ferrostatin-1 [21]. GPX4 also acts as defense enzyme against ferroptosis and is controlled by NRF2 [56,57]. Overexpression of GPX4 has been shown protective against ferroptosis [57,58]. Thus, our data showing significant suppression of GPX4 expression during fatal IOE infection suggest that virulent IOE evade host's antioxidant and anti-bacterial response and likely cause liver damage by inhibiting the expression of GPX4.

Notably, our data showed that inhibition of hepatic NRF2 activation during fatal IOE infection is due to decreased expression as well as defective translocation of NRF2 to the nucleus. Decreased translocation of NRF2 during IOE infection could be explained by stabilization of the KEAP1-NRF2 system. KEAP1 is an adaptor of the ubiquitin ligase complex that targets NRF2 for degradation. Activation of mammalian target of rapamycin complex 1 (mTORC1) under cellular stress has been shown to induce degradation of KEAP-1 by increasing its binding to phosphorylated p62, a selective autophagy protein that binds to ubiquitinylated proteins and damaged organelles targeting them to lysosomal compartments for degradation. This binding results in KEAP-1 degradation via autophagy, subsequent stabilization of NRF2, and translocation to nucleus [59]. We have previously showed that IOE infection of C57BL/6 mice triggers mTORC1 activation in the liver, which blocks autophagy induction and flux [7]. Thus, it is possible that IOE-mediated block of autophagy leads to inefficient degradation of KEAP-1, resulting in its stabilization and prevention of NRF2 translocation and activation. In support of this conclusion, our data suggests an impairment of mitophagy in IOE-infected HCs as marked by decreased expression of PINK1 and PARKIN, two ubiquitin molecules involved in mitophagy.

Several studies have shown that *Ehrlichia chaffeensis* inhibits apoptosis of macrophages, major target cells, as an immune evasion strategy to enable their intracellular survival and replication [60–62]. However, it is unclear whether similar mechanism happened in other target cells such as HCs. In humans, *Ehrlichia* is transmitted from skin to peripheral organs following tick bite and disseminate to peripheral organs. *Ehrlichia* is found to spread from cell to cell during early infection via the filopodia of macrophages. At late stages of infection, *Ehrlichia* induce rupture of host cell membrane and subsequent exit [63]. In humans, different strains of *Ehrlichia chaffeensis* exhibit distinct virulence traits, each delineated by specific genomic sequences [64]. However, this species causes abortive infection in immunocompetent mice and thus it is not ideal model to use for analysis of immunopathogenesis of fatal ehrlichiosis. Alternatively, the two strains used in this study EM and IOE are closely related to *E. chaffeensis* [63,64]. Infection of mice with EM and IOE causes nonfatal and fatal ehrlichiosis, respectively that recapitulate clinical, laboratory, immunological and pathological findings in patients with mild and fatal ehrlichiosis [7,8,65]. In this study, we found that virulent IOE, but not avirulent EM, induces oxidative stress in HCs to enable their dissemination and systemic infection. These data are consistent with similar immune evasion strategy employed by other bacterial pathogens [66–69]. Excessive generation of ROS also leads to metabolic reprogramming in host cells including altered protein and lipid metabolism [70,71]. Fatal *Ehrlichia* infection in

humans and mice correlates strongly with the development of liver steatosis. Our data suggest that oxidative stress, during fatal *Ehrlichia* infection may account for altered lipid metabolism and liver steatosis.

Our study demonstrated that inhibition of ER stress in IOE-infected HCs by TUDCA increase expression of NRF2, suggesting ER stress-mediated inhibition of NRF2 expression. Various metabolites or toxins secreted by bacteria can also modulate ER stress [72–74]. IOE express several tandem repeat proteins (TRPs), secreted proteins that are transport from phagosome where *Ehrlichia* reside to cytoplasm via type I secretion system [75]. These TRPs plays important roles in manipulating host cell processes, including cytoskeletal organization, cell signaling, transcriptional regulation, post-translational modifications, autophagy, and apoptosis[74–78]. Thus, it is possible that modulation of the unfolded protein response is mediated by TRPs proteins of virulent IOE. Additionally, the interaction of TRPs with KEAP-NRF2 system may account for observed decrease in NRF2 expression in IOE-infected HCs.

Finally, this study is highly translational as we show, for the first time, that treatment of cells and mice with D3T following fatal IOE infection abrogate deleterious caspase 11 activation, cell death/liver injury, oxidative stress, while improving bacterial clearance leading to protection against fatal disease. Thus, the D3T treatment of fatal ehrlichiosis was able to reverse all hallmarks of *Ehrlichia* pathogenesis. In summary, our research demonstrates that the protective effect of D3T in *Ehrlichia* infected HCs model is NRF2 mediated and D3T offers powerful antioxidant therapy that reprograms the cell in favor of *Ehrlichia* clearance and enhanced cell viability.

## Supporting information

**S1 Fig.** **(A)** Dot blot data analyzing intracellular expression of mitochondrial ROS in uninfected or infected HCs, cultured with or without D3T at 24hr p.i. using Mitosox Red. **(B)** Analysis of MFI of Mitosox Red using flow cytometry of same cells described in (**A**). **(C)** Immunofluorescence staining of uninfected and IOE-infected HCs and labeled with Mitotracker (Red) and DAPI (blue). Scale bar 20 μm. **(D)** Graph showing ratio of mitochondrial DNA and nuclear DNA measured by q-RT PCR. **(E)** Representative graph showing the cell proliferation assay results in uninfected, IOE-infected, and D3T-treated IOE-infected HCs. The assay measures the ability of these cells to proliferate or grow in number over 68h, and the graph shows how treatment with D3T affects the proliferation of IOE-infected HCs compared to uninfected or untreated IOE-infected or D3T treated-IOE infected cells. **(F)** Graph showing splenocytes in uninfected, D3T treated uninfected, IOE-infected, and D3T treated IOE-infected mice. **(G)** Quantification of TUNEL-positive kupffer cells and HCs/ 40x hpf in IOE-infected and D3T-treated-IOE-infected liver tissues on 7 day p.i.
(TIF)

**S1 Table. Table showing gene categories, protein, and gene symbols, analyzed proteins/ genes, gene name, function, and their upregulation and downregulation in condition: uninfected, EM infected and IOE infected.**
(DOCX)

**S2 Table. List of primers used in the study. The table has name of the genes, primer orientation, and their sequence.**
(DOCX)

**S1 Text. Other Material and Methods.**
(DOCX)

## Acknowledgments

We thank Dr. Yasuko Rikihisa (Ohio State University, Columbus, OH) for providing stock of *Ehrlichia*, *Ixodes ovatus Ehrlichia* (IOE) utilized in this study. We also thank the University of Illinois core facility's Transmission Electron Microscopy, confocal microscopy, and flow cytometry facilities. Special acknowledgment to Eileen Brister and Fei Mo for helping in the histological imaging.

## Author Contributions

**Conceptualization:** Abdeljabar El Andaloussi, Nahed Ismail.

**Data curation:** Aditya Kumar Sharma, Abdeljabar El Andaloussi, Nahed Ismail.

**Formal analysis:** Aditya Kumar Sharma, Nahed Ismail.

**Funding acquisition:** Nahed Ismail.

**Investigation:** Aditya Kumar Sharma, Abdeljabar El Andaloussi, Nahed Ismail.

**Methodology:** Aditya Kumar Sharma, Abdeljabar El Andaloussi, Nahed Ismail.

**Project administration:** Nahed Ismail.

**Writing – original draft:** Aditya Kumar Sharma.

**Writing – review & editing:** Nahed Ismail.

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
