## [Decision Letter · Decision Letter 0]

28 Jun 2023

Dear Dr Ismail,

Thank you very much for submitting your manuscript "Ehrlichia Inhibit NRF2 Activation in Hepatocytes to Trigger Oxidative Stress and Promote Liver Injury and Sepsis" for consideration at PLOS Pathogens. As with all papers reviewed by the journal, your manuscript was reviewed by members of the editorial board and by several independent reviewers. The reviewers appreciated the attention to an important topic. Based on the reviews, we are likely to accept this manuscript for publication, providing that you modify the manuscript according to the review recommendations.

Apologies for the delays that you have experienced in receiving a response, which was due to securing reviewers with the necessary background to appraise your work. Your manuscirpt has now been reviewed by two experts, who are generally positive about the quality and impact of the work. As you will see they raise important points, particularly about the immunoblots and cell fractionation and also the interpretation of the histological data.

Sincerely,

Richard D. Hayward

Guest Editor

PLOS Pathogens

David Skurnik

Section Editor

PLOS Pathogens

Kasturi Haldar

Editor-in-Chief

PLOS Pathogens

orcid.org/0000-0001-5065-158X

Michael Malim

Editor-in-Chief

PLOS Pathogens

orcid.org/0000-0002-7699-2064

Reviewer Comments (if any, and for reference):

Reviewer's Responses to Questions

**Part I - Summary**

Reviewer #1: The strengths of this manuscript include a well controlled set of studies demonstrating the role of NRF2 in the liver injury and sepsis mediated by Ehrlichia. Several in vivo and in vitro studies have been used to validate the findings.

While much of the transcirptional data is well controlled, there are some issues with the localization of NRF2 to nucleus/cytoplasm based on the immunoblots included. A cleanup of this issue will aid in clearly defining the role of NRF2 in ehrlichial pathogenesis. Histopathology slides are difficult to interpret unleass there is guidance on what to look for. Too many gene acronyms are used across the manuscript. A table indicating the list, their role and up/down regulation in uninfected, EM or IOE cells will aid in the understanding of this manuscript.

Reviewer #2: In the manuscript "Ehrlichia Inhibits NRF2 Activation in Hepatocytes to Trigger Oxidative Stress and Promote Liver Injury and Sepsis," the authors elucidate the mechanism by which a virulent strain of Ehrlichia affects hepatocytes upon infection. The important finding of this study is that the virulent strain of Ehrlichia (IOE) inhibits the expression and nuclear translocation of the NRF2 transcription factor in hepatocytes. NRF2 acts as a master regulator of redox homeostasis and metabolic balance during infection, protecting the host against oxidative stress by upregulating essential antioxidants. Thus, the inhibition of NRF2 expression and activity by IOE is proposed as a mechanism underlying severe inflammation and liver injury in ehrlichiosis. The authors also tested the possibility of confronting this mechanism with the application of 3H-1,2-dithiole-3-thione (D3T), a known potent inducer of NRF2 expression. The treatment of infected mice with D3T, as shown by the authors, prolonged animal survival and attenuated liver inflammation. Several underlying mechanisms, such as mitochondrial dysfunction, ER stress, and the accumulation of reactive oxygen species (ROS), were linked to the D3T/NRF2-regulated effects in vitro. Overall, the study presents novelty and is of broad interest to researchers working in the field of pathogen-host interactions. The in vivo studies are definitely a strength of this study, as well as the testing of a potential treatment option for the described severe condition. However, some clarification of the results, approaches, and method choices is still required.

**Part II – Major Issues: Key Experiments Required for Acceptance**

Reviewer #1: Sharma and others have demonstrated a novel mechanism by which highly infectious strain – Ixodes ovatus Ehrlichia (IOE) induces NRF2 activation in hepatocytes leading to inflammasome activation and liver injury using a murine model of fatal ehrlichiosis. The authors have used a number of different methods including a murine model of ehrlichiosis, histopathology, transcriptional and translational assays and a small molecule that targets NRF2 to establish the role of oxidative stress in the liver damage induced in fatal murine ehrlichiosis. The comparison between a fatal and a mildly virulent Ehrlichia muris (EM) is significant and the findings as described have strong experimental foundation. While these studies are critical to advance novel treatment options for human infectious monoculeosis, there are several issues that need to be addressed and will enhance the readability of this manuscript.

Major Issues:

1) Numerous genes -both chromosomal and mitochondrial- are described in this manuscript and is extremely difficult to follow due to differential regulation of these genes and extensive use of acronyms. It is critical to simplify by including a table with genes/proteins analyzed between IOE vs EM vs uninfected cellular/host responses in a table. The authors can use relative indicators (+, ++, +++ or -,--, --- ) with gene identities, their role in autophagy/oxidative stress response etc to simplify the data presented. The summary figure (8C) is also difficult to follow as many of the acronyms used in the figure are not expanded in the figure legend.

2) Figure 1. There is no clear description of this figure in the legend and in the description of the results. A-C is combined to state multiple inflammatory infiltrates, necrotic and apoptotic cells steatosis at day 7. Where can this be distinguished in IOE infected sections versus uninfected or EM infected sections?. Insets to distinguish these histological changes need to be marked or addressed individually in the results section. The reviewer was unbale to distinguish significant differences other than in 1B and it is unclear what that is supposed to reveal. More definition is needed both in the results section and in the figure legends and descriptions are very general and includes multiple statements with no clear representations. How is steatosis distinguished between IOE and EM.

3) Figure 2 ; A) There are two bands in the immunoblots with anti NFR2. Looks like the lower band is NFR2. If so, an arrow indicating that should be used. Is the upper band a cross reactive band or is it a different version of NFR2?. There is a major discrepancy in the levels of TXNRD1 levels being significantly elevated with IOE infection. There is no discussion of this deviation and also in line 288 in the discussion section TXNRD1 is indicated as downregulated. This aspect of Fig 2 need to be clarified and levels of TXNRD1 should be discussed in the context of this discrepancy.

Fig 2D. There are two bands detected with anti NRF2 antibodies. In the nuclear fractions, the top band is downregulated . Is this NRF2 translocated to the nucleus?. If so, it should be indicated by two arrows one for nuclear and the other cytoplasmic. There seems to some inconsistencies in the levels of nuclear and cytoplasmic levels in various immunoblots. The authors can remove this discrepancy by indicating the nuclear and cytoplasmic versions and quantify each version in all immunoblots….Multiple immunoblots with anti NRF2 antibodies show two bands or one band. Is the relative quantitation done for each band or for both bands together?. The identity of the bands used for quantitation should be indicated in the figure legend and arrows for nuclear and cytoplasmic versions included in each immunoblot.

Fig 3. The small molecule D3T used in the study has been used extensively in several studies. There are no relevant references (Line 146) to indicate that this small molecule mediates its function via NRF2. It is important to include a few refs as it appears that this small molecule is being validated for the first time even if it is true for study of oxidative stress response in murine ehrlichiosis. Fig 3A- is there a difference in nuclear and cytoplasmic levels of NRF2 in the presence of D3T?. How is the discrepancy between in vitro and in vivo data with regards to TXDR4 resolved?. Which is more relevant?. There needs to some discussion about this.

Fig 6B. The distortion/disorganization of ER appears to be the same between EM and iOE infected cells compared to uninfected cells. The authors may want to show another image or reword this sentence (Line 233). ER in uninfected cells is very clear while it appears to be the

same in Em/IOE infected cells. References needed after line 237/238.

Fig 7F. Is there a statistical difference in weight of liver between D3T treated and D3T treated plus IOE?.

Fig 8. Histological changes reflecting the pathology need to be marked/focused on the sections as it appears similar in all three sections to a reader.

A table listing all the chromosomal and mitochondrial genes/their functions with their noted effects will aid in a greater understanding of the proposed studies.

Reviewer #2: a) Results section:

1. Figure 2D: Nuclear translocation data are not supported by appropriate controls. Additional WB required to control for the purity of the fractions – addressing expression of nuclear protein (lamin, histone 3, PARP, any other) and cytoplasmic protein (GAPDH, Beta-tubulin, etc) on both fractions on one membrane for direct comparison of fractions. Ideally, total lysates should be loaded on the same membrane as well. Loading control is important, but it is not providing control for the proper fractions separation – possible contamination of cytosolic fraction with ruptured nuclear content or possible contamination of nuclear fraction by cytosolic proteins if nuclei were not properly washed before lysis.

APPROPRIATE SECTION describing the method should be added to the materials and methods part of the manuscript – how the separation was performed???

Could you comment on GPX4 size (below 17 kDa) as well as its function in the nucleus? Is it 22 kDa or shorter?

Do you have the total lysates to complement your mRNA expression data for GTX4 from Figure 2C by total protein expression level using WB on total lysate? (Like it as was done for total NRF2 on Figure 2A)?

2. Figure 7: It is a mess with subfigures description in the legend – should be checked and corrected by authors according to the presentation on the figure. Figure 1D, Figure 7B should have statistical significance and number of animals per group on the figure and figure legends.

b) Discussion section:

1. There are several repetitions of same conclusions that should be corrected, text requires English editing (example – lines 297-299; lines 310-312; lines 339-341)

2. References for statements required in lines 271-274. Line 277 – activation of caspase-11 is downstream inflammasome activation, not vice versa, like it reads now. Line – 288 – please list some processes and provide references. Statement on lines 317-318 requires reference. Could you comment if the mentioned strain (line 317) is virulent, how it is related to the strains that were used in current study. Line 329 – make sure that the journal allows to refer to unpublished data, usually it is not accepted. Line 332 – could you suggest if stability or complex with Keap-1 could be also affected.

c) Materials and methods:

1. Again – check that all methods were included, like for example the missing section on nuclear/cytosolic fractions isolation

2. Line 361 - Mice sex (male, female), time for acclimation before the experiment? Day-light cycle? There are certain protocols to describe the mouse experiments recommended by the journal as well that should be followed. Line 364 – decipher IOE strain again, provide the source for both strains that were used in the study. Line 366 – for D3T – what type of injection? Section Western blot (line 379) – as to the antibodies you should state if they were produced in mice, or rabbit, or rat, or goat, or etc.

3. RNA isolation and quantitative RT-PCR, bacterial burden determination – I request to provide all primers in main text or in supplementary file.

**Part III – Minor Issues: Editorial and Data Presentation Modifications**

Reviewer #1: It is all listed under major issues

Reviewer #2: 1) All genes titles should be correctly marked on graphs (italic, small letters for murine genes) and all throughout the text of the manuscript to separate where the authors address or describe protein expression, and where they examine mRNA expression of certain genes.

2) I would recommend deciphering the abbreviation EM early in the beginning of the results part (line 105)

3) Section (after line 218) should contain brief explanation for the choice of genes addressed in the study and abbreviations should be also explained. Why these genes? Any references that could be provided to explain the choice?

4) Lines 147-248: Is it only about expression or also the stability of NRF2 protein – disconnection from Keap1? Both? Is it possible to address?

5) Figure 3B-D – what was the concentration of D3T selected for these assays – please note in the figure or figure legend. Remove mark (µm) from Figure 4C,D or actually write the concentration used - 2.5 µM.

6) Have you tried staining Nrf2 in liver sections of infected mice? Is it possible to compare for the expression level visually, by comparing the staining intensity in levers of uninfected, EM- and IOE-infected mice by fluorescent staining of sections for example?

7) Figure 5C, 6D – molecular size should be noted on WBs.

8) Is it correct to state “NRF2 signllaing pathways” (line 198, line 283)? Because NRF2 is not activating directly the signaling events, but rather activate certain gene expression profile as a transcription factor.

9) Line 429 – not clear, should be edited – “treated to D3T treatment”? Line 431 – specify “a large number” by providing the actual range for the number.

PLOS authors have the option to publish the peer review history of their article (what does this mean?). If published, this will include your full peer review and any attached files.

Reviewer #1: No

Reviewer #2: No

Figure Files:

Data Requirements:

Reproducibility:

References:

---

## [Decision Letter · Decision Letter 1]

16 Oct 2023

Dear Dr Ismail,

Thank you very much for submitting your manuscript "Evasion of Host Antioxidative Response via Disruption of NRF2 Signaling in Fatal Ehrlichia-Induced Liver Injury" for consideration at PLOS Pathogens. As with all papers reviewed by the journal, your manuscript was reviewed by members of the editorial board and by several independent reviewers. The reviewers appreciated the attention to an important topic. Based on the reviews, we are likely to accept this manuscript for publication, providing that you modify the manuscript according to the review recommendations.

Your revised manuscript hs been reviewed by one of the original referees. They raise some additional points surrounding the clarity of the data presentation. These are important and should be attended to.

Sincerely,

Richard D. Hayward

Guest Editor

PLOS Pathogens

David Skurnik

Section Editor

PLOS Pathogens

Kasturi Haldar

Editor-in-Chief

PLOS Pathogens

orcid.org/0000-0001-5065-158X

Michael Malim

Editor-in-Chief

PLOS Pathogens

orcid.org/0000-0002-7699-2064

Your revised manuscript hs been reviewed by one of the original referees. They raise some additional points surrounding the clarity of the data presentation. These are important and should be attended to.

Reviewer Comments (if any, and for reference):

Reviewer's Responses to Questions

**Part I - Summary**

Reviewer #2: The manuscript was significantly improved and many issues were nicely clarified after the revision, however, there are still some changes that should be incorporated to improve the presentation of the results.

**Part II – Major Issues: Key Experiments Required for Acceptance**

Reviewer #2: (No Response)

**Part III – Minor Issues: Editorial and Data Presentation Modifications**

Reviewer #2: Here are some minor comments about the modifications that should be made to improve the clarity of the manuscript:

Figures:

1) I would highly suggest to change the color of arrows in Figure 1A,B - black arrows are almost not visible in printed text

2) Figure 2 - mention the meaning of T, N,C abbreviations in the brackets in the figure legend - same for Figure 3 B

3) Figure 3 title - "increase in NRF2" - expression? activity? stability? - be specific

4) Figure 5, 6 and 7 - it would be important to add molecular weight values for the blots presented on these images as you have done for the blots in other Figures. You have mentioned in the answer to the reviewer that it was done, but it was not done in reality.

5) Figure 6D - you should mark in the figure legend if the protein samples were run in the same gel and quantified from the same membrane

Text and figures - use uniformly the term cytoplasmic fraction - you use both cytosolic and cytoplasmic terms in the text and in the figures and legends.

Paper text:

1) Line 162 - There is a significant decrease of NRF2 expression in total lysates along with both nuclear and cytoplasmic fractions (Figure 2D). It is showing the effect of IOE on the NRF2 stability at the first place. In line 163 you rather put forward translocation and expression, however, there are no mRNA expression data for Nrf2 gene in this figure. This part of the text and conclusions should be edited.

2) Line 310 - edit the text, and it is a general comment for all newly inserted sections

3) Line 318-319 - I would suggest to change to: In contrast, treatment of IOE infected mice with D3T not only decreased the activation of caspase-11 in liver tissue ... - isn't it more correct?

4) Discussion - I would suggest moving and combining new text from line 345-347 to the line 356

PLOS authors have the option to publish the peer review history of their article (what does this mean?). If published, this will include your full peer review and any attached files.

Reviewer #2: **Yes: **Maria Yurchenko, PhD, Researcher, CEMIR, IKOM, NTNU, Trondheim, Norway

Figure Files:

Data Requirements:

Reproducibility:

References:

---

## [Editor Report · Decision Letter 2]

30 Oct 2023

Dear Dr Ismail,

We are pleased to inform you that your manuscript 'Evasion of Host Antioxidative Response via Disruption of NRF2 Signaling in Fatal Ehrlichia-Induced Liver Injury' has been provisionally accepted for publication in PLOS Pathogens.

Best regards,

Richard D. Hayward

Guest Editor

PLOS Pathogens

David Skurnik

Section Editor

PLOS Pathogens

Kasturi Haldar

Editor-in-Chief

PLOS Pathogens

orcid.org/0000-0001-5065-158X

Michael Malim

Editor-in-Chief

PLOS Pathogens

orcid.org/0000-0002-7699-2064

Thank you for your attention to the minor comments raised by the reviewer.
---

## [Editor Report · Acceptance letter]

6 Nov 2023

Dear Dr Ismail,

We are delighted to inform you that your manuscript, "Evasion of Host Antioxidative Response via Disruption of NRF2 Signaling in Fatal <i>Ehrlichia<i>-Induced Liver Injury," has been formally accepted for publication in PLOS Pathogens.

Best regards,

Kasturi Haldar

Editor-in-Chief

PLOS Pathogens

orcid.org/0000-0001-5065-158X

Michael Malim

Editor-in-Chief

PLOS Pathogens

orcid.org/0000-0002-7699-2064